



# Earth system modeling of mercury using CESM2: part 1. atmospheric model CAM6-Chem/Hg v1.0

Peng Zhang[1], Yanxu Zhang[1]

[1]School of Atmospheric Science, Nanjing University, Nanjing, China

*Correspondence to*: Yanxu Zhang (zhangyx@nju.edu.cn)

**Abstract.** Most global atmospheric mercury models use offline and reanalyzed meteorological fields, which has the advantages of higher accuracy and lower computational cost compared to online models, but they have limited capacity in predicting the future. Here, we use an atmospheric component with tropospheric and stratospheric chemistry (CAM6-Chem) of the state-of-the-art global climate model CESM2, adding new species of mercury and simulating atmospheric mercury cycling. Our results

show that the newly developed online model is able to simulate the observed spatial distribution of total gaseous mercury (TGM) in both polluted and non-polluted regions with high correlation coefficients in East Asia ($r = 0.67$) and North America ($r = 0.57$). The calculated lifetime of TGM against deposition is 5.3 months and reproduces the observed interhemispheric gradient of TGM with a peak value at northern mid-latitudes. Our model reproduces the observed spatial distribution of $Hg^{II}$ wet deposition over North America ($r = 0.80$) and captures the magnitude of maximum in the Florida Peninsula. The simulated

wet deposition fluxes in East Asia present a spatial distribution pattern of low in the northwest and high in the southeast.  The online model is in line with the observed seasonal variations of TGM at northern mid-latitudes as well as the Southern Hemisphere, which shows lower amplitude. We further go into the factors that affect the seasonal variations of atmospheric mercury and find that both $Hg^0$ dry deposition and $Hg^{II}$ dry/wet depositions contribute to it.

## 1 Introduction

Mercury (Hg) is a global pollutant that can be transported and exchanged among the atmosphere, geosphere, hydrosphere, cryosphere, and biosphere (Selin, 2009; Driscoll et al., 2013). Although the anthropogenic emissions are mainly to the atmosphere, most human exposure for the general population to its toxic form (methylmercury) is mainly through consumption of seafood and rice (Zhang et al., 2010; Mahaffey et al., 2011; Zhang et al., 2021). Understanding the relationship between anthropogenic emissions and human exposure requires a holistic view of the global Hg cycle, and an Earth system model that

integrates the multiple spheres is required. As the first step of this effort, here we develop a new online atmospheric Hg model (CAM6-Chem/Hg v1.0, hereinafter referred to "CAM6-Chem/Hg") within the Community Earth System Model version 2 (CESM2).

One advantage of our online model is that the concentrations of Hg oxidants are calculated online. There are three main stable forms of atmospheric mercury: gaseous elemental mercury ($Hg^0$), reactive gaseous mercury ($Hg^2$), and particulate bound



mercury ($Hg^P$). Up to now, several $Hg^0$ oxidants have been proposed and the debate about which is the dominant oxidant has focused on $O_3$, OH radical or halogen species (Si and Ariya, 2018; Lyman et al., 2020). In the previous global modeling studies, using Br as the only oxidation pathway (Holmes et al., 2010; Amos et al., 2012; Horowitz et al., 2017) or oxidation by $O_3$/OH (Selin et al., 2007; De Simone et al., 2014; Pacyna et al., 2016) can both get reasonable results. Nevertheless, recent studies seemed to indicate that more complex chemistry and multiple oxidants of $Hg^0$ exist in the atmosphere (Weiss-Penzias et al.,

2015; Travnikov et al., 2017; Si and Ariya, 2018; Lyman et al., 2020; Shah et al., 2021). Previous models often use archived monthly mean concentrations for these oxidants (e.g., Seigneur et al., 2001; Durnford et al., 2012; Shah et al., 2021), which neglects the diurnal variation of these species. Similar to the meteorological data, using offline oxidants also lacks the predicting capacity for the oxidizing power of the atmosphere under future emission and climate conditions.

        The other advantage of our model is that the meteorological fields are calculated online and coupled with chemistry. So

far, most of the global atmospheric Hg models are offline models, such as GLEMOS (Travnikov and Ilyin, 2009), GEOS-Chem-Hg (Selin et al., 2007), CAM-Chem/Hg (Lei et al., 2013), and CTM-Hg (Seigneur et al., 2001). These offline chemical transport models (CTMs) are driven by the meteorological reanalysis data which are derived from external numerical weather prediction (NWP) or climate models. Advantages of the assimilated meteorological data include accuracy and relatively low computational cost (El-Harbawi, 2013). However, one drawback for the atmospheric chemistry models that rely on it is the

lack of future data. The inputted meteorological data often need to be interpolated in time and space and the physical parameterizations (e.g., advection and convection) are commonly different between CTMs and NWP, leading to inconsistency in the modeling results (Grell and Baklanov, 2011). The interaction between meteorology and atmospheric chemistry is also often not considered in these models. In our model, no interpolation in time or space for the meteorological data is needed, the same numerical schemes make the online model more consistent than the offline model, and the impact of meteorology and

chemistry feedbacks can also be taken into account (Baklanov et al., 2014). Furthermore, the online Hg model is of great significance to forecast or simulate the impacts of future climate change on Hg.

        We add atmospheric Hg species in the Community Atmosphere Model version 6 with chemistry (CAM6-chem), which does not include any Hg species. The related processes such as anthropogenic and natural emissions, dry and wet deposition, and redox chemistry of Hg are also included. The CAM-chem model has been widely used for simulations of global

tropospheric and stratospheric atmospheric composition, such as OH (Wang et al., 2020), $O_3$ (Emmons et al., 2020), and halogens (Fernandez et al., 2019). Lei et al. (2013) also developed an offline CAM-Chem/Hg model under the Community Climate System Model version 3 (CCSM3), which is the predecessor and has been greatly improved by the CESM2 used in this study. We run the CAM6-chem/Hg model in the free-running configuration, which provides a platform for online modeling of different atmospheric Hg species. The main objective of this study is to test the performance of the model driven by online

meteorology data for Hg by extensively comparing the model results with available observations worldwide.





## 2 Model description

### 2.1 CAM6-Chem model

The CAM6-Chem, based on the Community Atmosphere Model version 6 (CAM6), is the atmospheric component with chemistry of the CESM2 (Emmons et al., 2020). The CESM2 is an open-source community coupled model, consisting of seven major prognostic components: atmosphere, land, land-ice, ocean, sea-ice, river runoff, and wave (Danabasoglu et al., 2020). The CAM6 uses the same Finite Volume (FV) dynamic core (Lin and Rood, 1997) as its predecessors CCSM4 and CESM1 but with a variety of modifications (see Danabasoglu et al., 2020 for more details). The CAM6 model version we used here has a horizontal resolution of 0.9° in latitude by 1.25° in longitude, with 32 vertical levels from the surface to 2 hPa (about 45 km).

The atmospheric chemical mechanism of CAM6-Chem is based on the Model of Ozone and Related chemical Tracers-Tropospheric and Stratospheric (MOZART-TS1) chemistry, which contains 221 solution species and 528 chemical reactions (including 405 kinetic and 123 photolysis reactions) (Emmons et al., 2020). The chemical species within the TS1 mechanism are similar to that of another chemistry configuration of CAM6 (i.e., WACCM6), including $O_x$, $HO_x$, $NO_x$, $ClO_x$, and $BrO_x$ chemical families, along with $CH_4$ and its degradation products (Gettelman et al., 2019). Aerosols in CAM6-Chem are represented using the four-mode version of the Modal Aerosol Model (MAM4), including sulfate, black carbon, primary organic matter, secondary organic aerosols, sea salt, and mineral dust (Liu et al., 2016). Secondary organic aerosols are treated using a volatility basis set (VBS) scheme, which is described in detail by Tilmes et al. (2019).

The anthropogenic emissions in the CAM6-Chem for 1750–2014 are based on the Coupled Model Intercomparison Project Phase 6 (CMIP6) inventories, provided by the Community Emissions Data System (CEDS) (Hoesly et al., 2018). We use the anthropogenic emissions of 2010 in this study. Biomass burning emissions for 1750–2015 are based on van Marle et al. (2017) and are all surface emissions (i.e., without any plume-rise or vertical emissions). In addition, CAM6-chem is coupled to the interactive Community Land Model (CLM5), which calculates the biogenic emissions online based on the Model of Emissions of Gases and Aerosols from Nature (MEGAN-v2.1) (Guenther et al., 2012).

The CAM6-Chem calculates the dry deposition velocity of gas-phase compounds following the resistance-in-series scheme of Wesely (1989) through the coupled CLM5. CLM5 provides a parameterized scheme based on five seasonal categories and 11 land use types for the estimation of surface resistance, which ensures that the changes in climate, land cover, and land use can affect the deposition process (Lamarque et al., 2012). The wet deposition of soluble gas-phase compounds in CAM6-Chem is based on the scheme of Neu and Prather (2012), consisting of two processes: in-cloud and below-cloud scavenging.

### 2.2 Mercury emission

Both anthropogenic and natural Hg emissions are included in CAM6-Chem/Hg model (Figure 1). The anthropogenic emissions are based on an improved 2010 global inventory developed by Zhang et al. (2016), which considered the release of





Hg from commercial products and emission controls on coal combustion. The total anthropogenic emissions are 2275 Mg a$^{-1}$, of which 1470 Mg a$^{-1}$ and 805 Mg a$^{-1}$ are for Hg$^0$ and Hg$^{II}$ respectively.

The natural emissions are derived from the average of a 5-year simulation in GEOS-Chem, including geogenic, biomass burning, soil, snow, and vegetation emissions. The simulation is conducted at 4° × 5° horizontal resolution and then mapped to 0.9° × 1.25° for CAM6-Chem. Since the ocean model (POP2) of the CESM2 used in CAM6-Chem/Hg is a data component, the exchange process of Hg$^0$ between atmosphere and ocean is replaced with a net emission from ocean. This net ocean emission is about 3200 Mg a$^{-1}$ and falls in the range of 2840 to 3710 Mg a$^{-1}$ indicated by a recent online air-sea coupling model

research (Zhang et al., 2019). The natural emission from land is about 1500 Mg a$^{-1}$, which is similar to the previous estimation used in GEOS-Chem (Horowitz et al., 2017). Future work will improve the calculation of the fluxes by introducing Hg tracer into other models (e.g., CLM5 and POP2) and coupling with CAM6-Chem/Hg online within the CESM2/Hg.

### 2.3 Mercury chemistry

Possible oxidants of Hg$^0$ include O$_3$/OH, Br, or other halogens. It is noted that neither oxidation by Br or O$_3$/OH alone in

the model could reproduce the global or regional observations (Wang et al., 2014; Weiss-Penzias et al., 2015; Ye et al., 2016; Bieser et al., 2017; Gencarelli et al., 2017; Travnikov et al., 2017). Therefore, We adopt multiple pathways including both Br and O$_3$/OH as the oxidants of Hg$^0$. The Hg$^0$ oxidation by Br atoms is considered as a two-step process with rates following Horowitz et al., (2017). The oxidation by O$_3$/OH adopts the reaction rates used in Lei et al. (2013), except that the product is assumed to be Hg$^{II}$ in our simulation. Different from the Hg$^{II}$ 50/50 partitioning used in the previous studies (Holmes et al.,

2010; Lei et al., 2013), we adopt an empirical relationship introduced by Amos et al. (2012) to deal with the gas-particle partitioning of Hg$^{II}$, considering the influence of fine particulate matter (PM2.5) and temperature. Several possible Hg$^{II}$ reduction pathways have been proposed (Si and Ariya, 2018) and we adopt the aqueous photoreduction of Hg$^{II}$-organic complexes used in Horowitz et al. (2017). The best match with the available observations can be obtained by adjusting the photoreduction rate coefficient.

The representation of the main oxidants (e.g., O$_3$ and OH) of Hg$^0$ have been greatly improved and are more comprehensive in the CAM6-Chem, comparing with its predecessors CESM or CCSM (Lamarque et al., 2012; Emmons et al., 2020). The CAM6-Chem contains the sources of Br from the photodecomposition of organobromines (CHBr$_3$, CH$_2$Br$_2$, CH$_3$Br), the homogeneous and heterogeneous bromine chemistry reactions, as well as the chlorine chemistry (for more details see TS1 mechanism in Emmons et al., 2020). Our model takes advantage of these improvements and couples the atmospheric chemistry

of Hg with its oxidants for the first time.

### 2.4 Mercury deposition

The removal of Hg from the atmosphere includes wet and dry deposition processes. Considering the differences in physical and chemical properties of atmospheric Hg species, different approaches are used to simulate deposition in CAM6-Chem/Hg. For Hg$^0$ and Hg$^2$, we adopt the default schemes used in CAM6-Chem for the dry and wet deposition of gas-phase





compounds (see Section 2.1). The dry deposition rates of $Hg^0$ and $Hg^2$ are calculated online in the CLM5 model based on the land use types. The wet removal of $Hg^2$ includes both in-cloud and below-cloud processes. In addition, the uptake and deposition of $Hg^2$ by sea-salt aerosol are parameterized following Holmes et al. (2010) as a first-order loss rate in the marine boundary layer (MBL). For $Hg^P$, its treatment for dry and wet deposition is the same as other aerosol species in CAM6-Chem.

### 2.5 Model run

In this work, a 4-year (2010–2013) free-running simulation is conducted in CAM6-Chem/Hg, with specified sea surface temperature. We use the initial year (i.e., 2010) as a spin-up to minimize the impact of initial conditions, and the average of the following three years (i.e., 2011–2013) for analysis (unless explicitly stated). The model's initial state is taken from the output of the GEOS-Chem model (Horowitz et al., 2017) and linearly interpolated to the CAM6-Chem model grid.

## 3 Model evaluation

### 3.1 Global atmospheric Hg budget

Figure 1 shows the global budget of Hg derived from our CAM6-Chem/Hg simulation, including the reactions with three major oxidants (Br, OH, and $O_3$) and the photoreduction rates. The oxidation with OH radicals is an important pathway when considering the proportion of oxidation rates, consisting with previous studies (Selin et al., 2007; Shah et al., 2021). Indeed, recent research has found that the main products $HOHg^I$ and $Hg(OH)_2$ formed by the oxidation of $Hg^0$ by OH are relatively

stable (Dibble et al., 2020). The reservoir of global atmospheric Hg is 3896 Mg, of which $Hg^0$ and $Hg^{II}$ are 3566 Mg and 330 Mg, respectively (for $Hg^{II}$, 122 Mg and 208 Mg in the gaseous and particulate phases, respectively). The global total Hg emissions from all sources are about 7000 Mg a$^{-1}$, two-thirds of these are natural or legacy emissions. In this study, the rapid re-emission of deposited $Hg^0$ is included in the land emission and the $Hg^0$ dry deposition to the ocean is replaced with the net $Hg^0$ evasion. The lifetime of total gaseous mercury (TGM = $Hg^0$ + $Hg^2$) against deposition is 5.3 months in our simulation,

which is in accordance with the estimation in the GEOS-Chem model (Holmes et al., 2010; Horowitz et al., 2017).

The modeled vertical distributions of the annual zonal mean mixing ratios of $Hg^0$ and $Hg^{II}$ are shown in Figure 2. $Hg^0$ is the dominant chemical species in the troposphere and is higher in the Northern Hemisphere (NH) than in the Southern Hemisphere (SH) due to anthropogenic emissions. The simulated $Hg^0$ concentration is relatively uniform in the troposphere but decreases rapidly around the tropopause. The simulated $Hg^{II}$ increases with altitude and is more abundant in the stratosphere.

These vertical distributions of $Hg^0$ and $Hg^{II}$ are consistent with previous model studies (Holmes et al., 2010; Horowitz et al., 2017) and available aircraft measurements (Talbot et al., 2008; Lyman and Jaffe, 2012; Slemr et al., 2018).



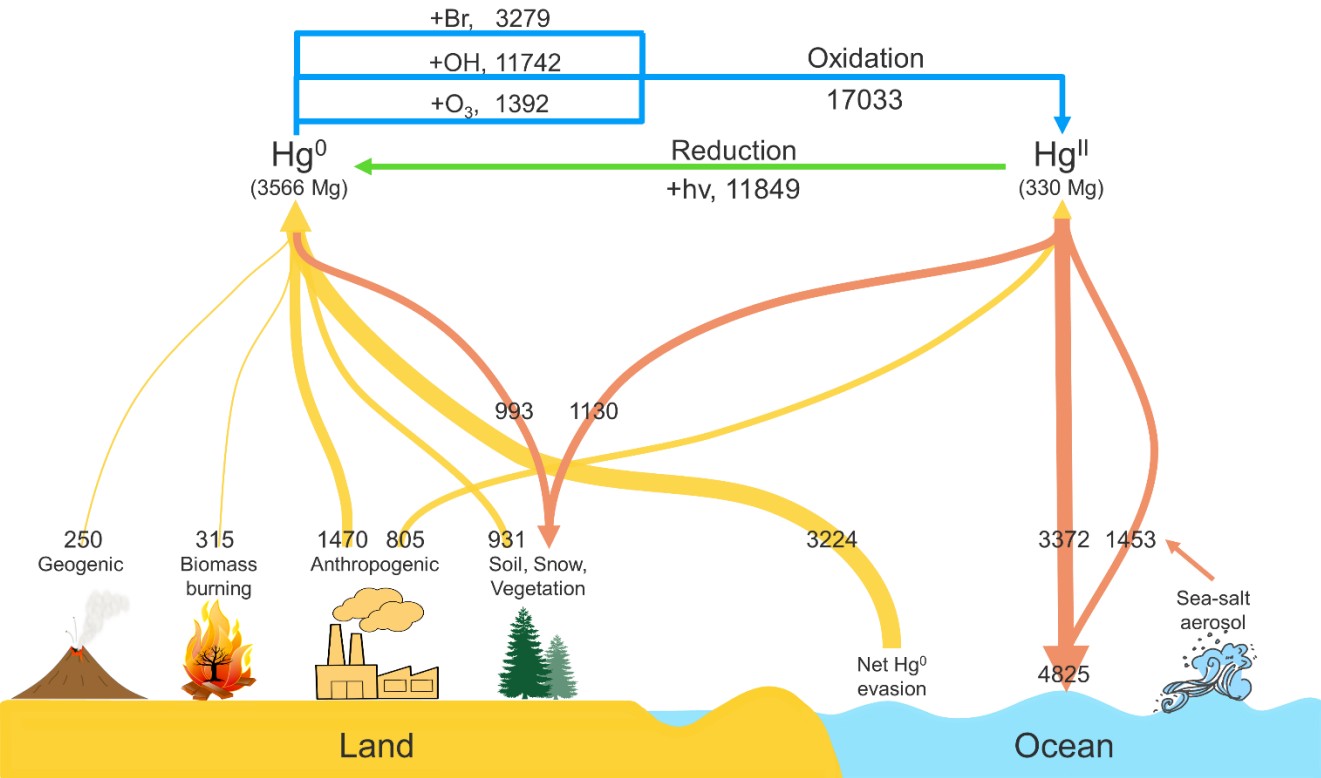

**Figure 1.** Global budget of atmospheric mercury in CAM6-Chem/Hg. Only the reaction rates of three major oxidants are shown. Fluxes in Mg a$^{-1}$. The thickness of the arrows represents the relative contribution.


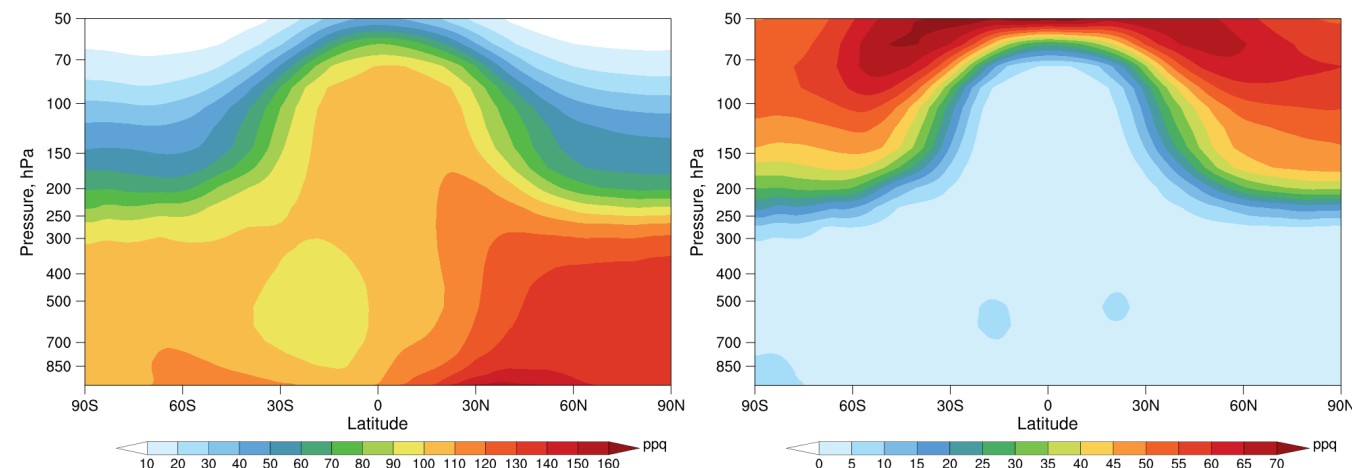

**Figure 2.** Annual (2011–2013) zonal mean concentrations of Hg$^0$ (left) and Hg$^{II}$ (right) in CAM6-Chem/Hg.

## 3.2 Atmospheric Hg concentrations

The global distribution of surface TGM concentrations is shown in Figure 3 with our simulation compared to the available
ground-based observations during 2009–2015 (except for some observations in East Asia that are out of this time range, see





Table S1 for detailed description). In general, the mean and standard deviation at the 92 land sites are $1.78 \pm 1.02$ ng m$^{-3}$ for the model, agreeing well with the observations ($1.75 \pm 0.86$ ng m$^{-3}$). Our model also reproduces the global distribution of TGM well with high correlation coefficients ($r = 0.83$).

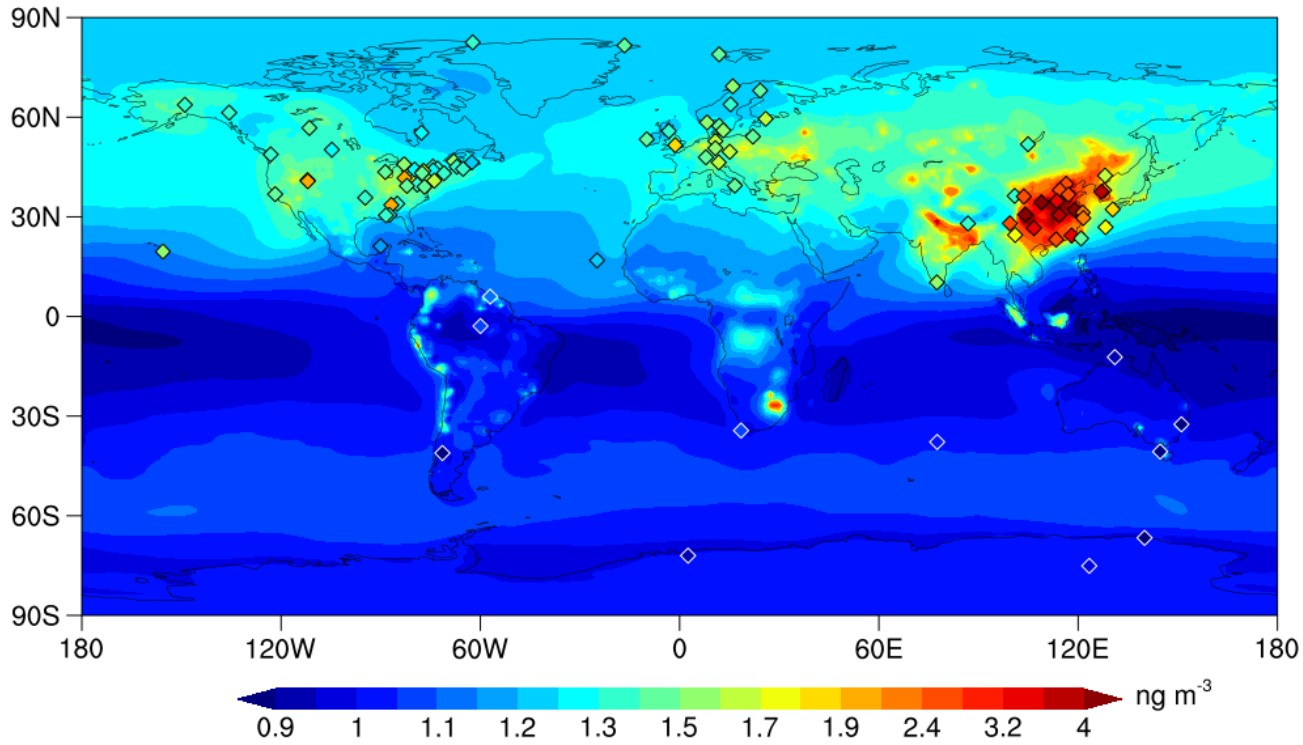

**Figure 3.** Global distribution of surface total gaseous mercury (TGM) concentrations. Model results (background) are annual average of 2011–2013. Ground-based observations (rhombuses) are for 2009–2015 except for some observations in East Asia (see Table S1 in the Supplement for detailed description). The rhombuses with white frames are concentrations below 1.2 ng m$^{-3}$. Note the scale of the color bar is non-uniform.

The modeled global mean of surface TGM concentrations is 1.14 ng m$^{-3}$ with a higher value in the NH (1.27 ng m$^{-3}$) than in the SH (1.02 ng m$^{-3}$), indicating a significant cross-hemisphere gradient (Figure 4). Overall, the CAM6-Chem/Hg can capture the peak values in the northern mid-latitudes and the lower ones in the SH. A gradual increasing trend from south to north is in line with the limited land-based observations in the tropics (Figure 4). This simulated interhemispheric gradient of TGM also agrees with observational studies in the marine boundary layer (Soerensen et al., 2010a; Sprovieri et al., 2010;

Sprovieri et al., 2016) and previous modeling studies (Lamborg et al., 2002; Selin et al., 2007; Holmes et al., 2010; Travnikov et al., 2017). We attribute this interhemispheric gradient to the large disparity in Hg emissions in the two hemispheres. In CAM6-Chem/Hg, the total anthropogenic emissions in the NH are about 7-fold higher than that in the SH and are 1.5-fold for total emissions (natural emissions in the SH partially offset the large difference in emissions).





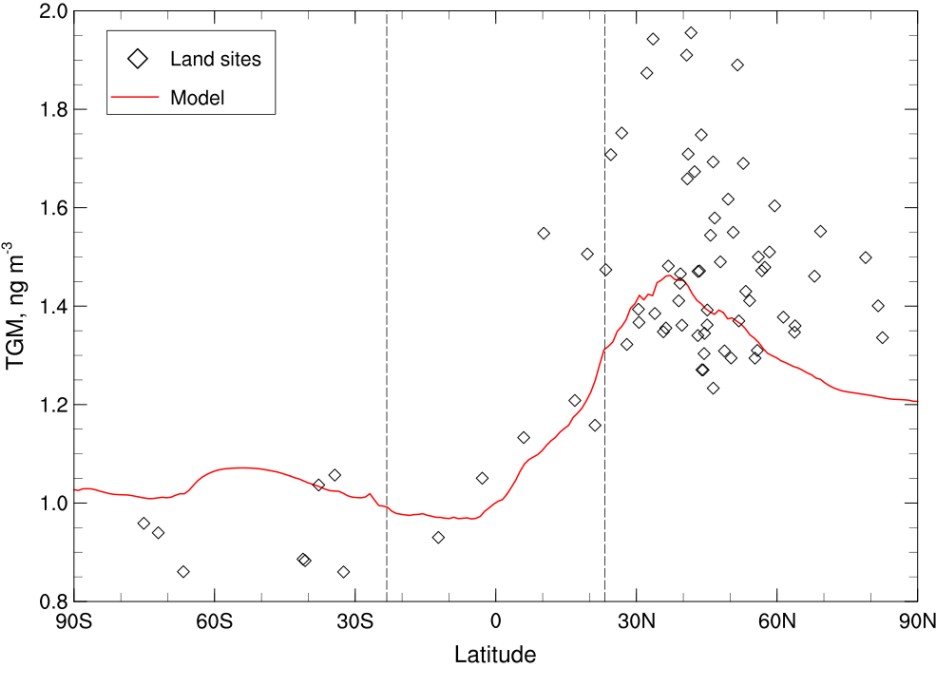

**Figure 4.** The latitudinal variation of surface TGM concentrations. The red line is the modeled zonal mean result for 2011–2013. The two black dashed lines represent the locations of the Tropic of Capricorn and Cancer. The observations (rhombuses) at land sites only include part of the values (< 2 ng m$^{-3}$) as shown in Fig. 3.

The standard deviations of TGM concentrations are large for both model and observations, indicating large regional variability, especially between polluted and background areas. Figure 5 shows the modeled results in three regions with elevated TGM concentrations: East Asia (10°–55°N, 70°E–140°E), West Europe (40°N–70°N, 10°W–25°E) and North America (20°N–60°N, 60°W–125°W). East Asia is one of the regions most contaminated by Hg in the world (Sprovieri et al., 2016). The observed mean TGM concentrations at the available 26 land sites are 2.69 ± 1.16 ng m$^{-3}$. The high standard deviation reflects the large spatial variability within East Asia. Eight sites are higher than 3 ng m$^{-3}$ and three are above 4 ng m$^{-3}$, including Xi'an (5.66 ng m$^{-3}$, Xu et al., 2017), Nanjing (4.98 ng m$^{-3}$, our own data), and Chengdu (4.56 ng m$^{-3}$, Fu et al., 2021) in China. These high-concentration sites are usually located in urban or suburban regions which are close to the large anthropogenic emission sources. Our model agrees with the observations quite well (2.97 ± 1.28 ng m$^{-3}$) with a spatial correlation coefficient ($r$) of 0.67. The observed TGM concentrations in West Europe are lower (1.53 ± 0.15 ng m$^{-3}$, n = 15 sites) than in East Asia. We find the model (1.31 ± 0.09 ng m$^{-3}$) slightly underestimates the observations ($r$ = 0.22), but the model simulates high values in central and eastern regions, which is consistent with the observations and high emissions in these areas (Pacyna et al., 2001, 2006). In North America, the observed TGM concentrations are 1.47 ± 0.20 ng m$^{-3}$ (n = 30), similar to the TGM level in West Europe. The model (1.43 ± 0.15 ng m$^{-3}$) also agrees well with the observations ($r$ = 0.57). The modeled TGM shows relatively higher concentrations in the eastern and western United States but relatively uniform and low in the central area, consistent with the spatial distribution of anthropogenic emissions of Hg (the scatter diagrams of





comparison between simulation and observations are shown in Fig. S1 in the Supplement). The agreement between observations and model results in these areas implies that our model performs well in polluted regions. In addition, the modeled TGM concentrations in north-eastern India are also high (left panel in Figure 5), calling for more observations to constrain the model.

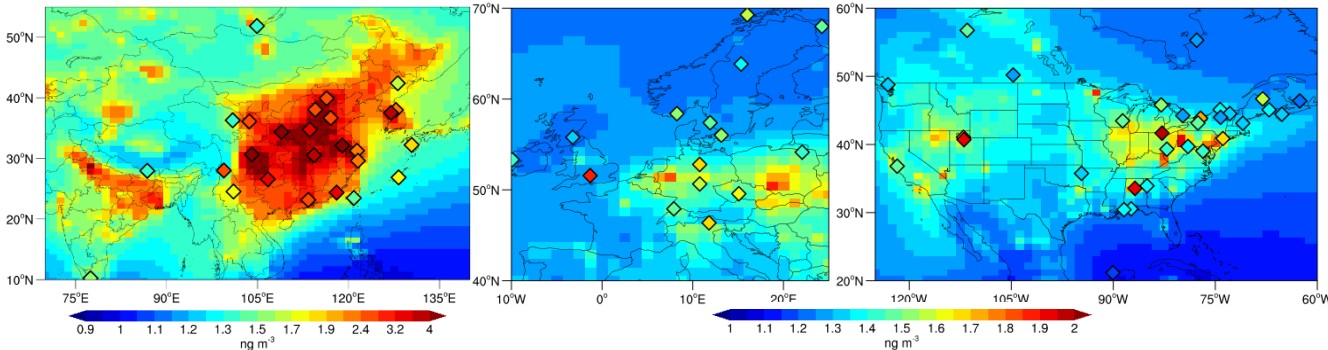

**Figure 5.** Regional distribution of surface TGM concentrations in East Asia, West Europe, and North America. Ground-based observations (rhombuses) are the same as Fig. 3. Note two different color bars are used to depict regional variations.

## 3.3 Atmospheric Hg depositions

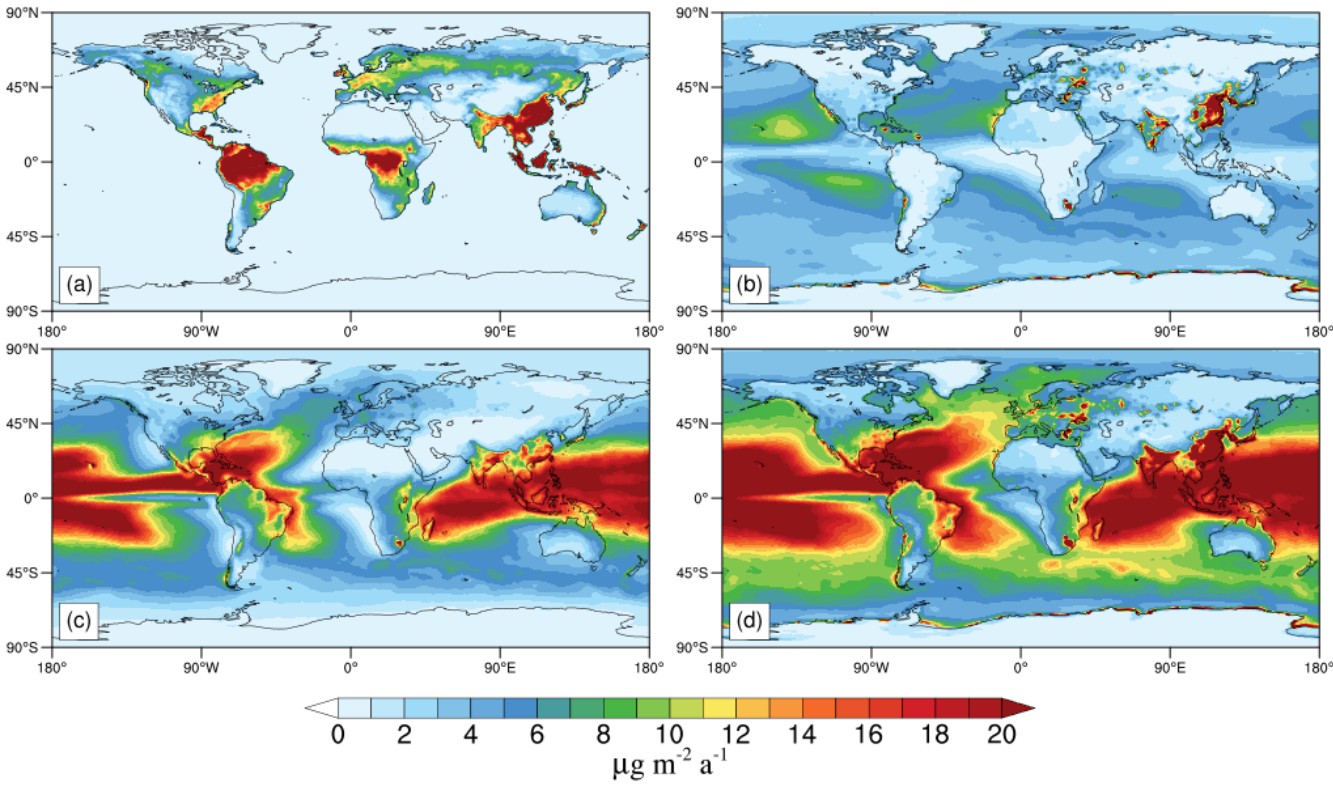

**Figure 6.** Global distribution of annual mean Hg deposition fluxes during 2011–2013 in CAM6-Chem/Hg: (a) $Hg^0$ dry deposition, (b) $Hg^{II}$
dry deposition (including sea-salt uptake), (c) $Hg^{II}$ wet deposition, and (d) $Hg^{II}$ total deposition.



Figure 6a shows the distribution of Hg$^0$ dry deposition worldwide (the deposition to the ocean is considered as a net re-emission from the ocean, i.e., gross evasion minus deposition). The annual global dry deposition flux of Hg$^0$ over land is 993 Mg a$^{-1}$ and more than half is distributed in the tropics (23.26°S–23.26°N). This Hg$^0$ deposition flux is slightly smaller than

previous GEOS-Chem models (1200–1600 Mg a$^{-1}$, Selin et al., 2008; Holmes et al., 2010; Horowitz et al., 2017) due to different parameter settings (e.g., grid resolution, land types, and meteorological data) between these models. The dry deposition flux is based on the dry deposition velocity and the local Hg$^0$ concentration. Fluxes are higher over northern South America, central Africa, Southeast Asia, and East Asia. The high flux in East Asia is driven by the high Hg$^0$ concentration, whereas the Hg$^0$ concentrations over the other three regions are relatively low (see Figure 3). As described in section 2.1, the

dry deposition velocities are affected by the leaf area index (LAI), which depends on seasons and land use types, as modeled by the CLM5. The major land use type of these three regions is tropical rainforests, which leads to a high dry deposition velocity and subsequently high dry deposition flux.

Figures 6b and 6c show the global distribution of dry and wet annual deposition flux of Hg$^{II}$ respectively (the uptake of Hg$^{II}$ by sea-salt aerosol is included in dry deposition). The total Hg$^{II}$ deposition is 5954 Mg a$^{-1}$ (Figure 6d), of which the

proportions of dry and wet deposition are 9% and 67% respectively, compared to 16% and 61% in Holmes et al. (2010). The rest is the uptake by sea-salt aerosol (1453 Mg a$^{-1}$), which has been proved to be an important sink for Hg$^{II}$ in the MBL in previous researches (Holmes et al., 2009, 2010; Malcolm et al., 2009). Hg$^{II}$ deposited to the ocean accounts for 81% of the global total deposition, while the land takes the remaining 19%. These proportions are consistent with Horowitz et al. (2017), but our model shows a higher flux in low latitudes. The global total Hg$^{II}$ deposition to tropical oceans (30°S–30°N) in our

simulation is 3328 Mg a$^{-1}$, higher than 2800 Mg a$^{-1}$ in Horowitz et al. (2017). This is partly due to the addition of OH/O$_3$ oxidation mechanisms. The OH concentration is the greatest at low latitudes in the lower to the middle troposphere (Crutzen and Zimmermann, 1991; Spivakovsky et al., 2000; Lelieveld et al., 2016), which causes more Hg$^0$ oxidation to Hg$^{II}$ and the frequent deep convective precipitation promotes the wet removal of Hg$^{II}$ in the tropics. Another reason for the difference could be the different Br concentrations. As the debromination of sea salt aerosol is not included in CAM6-Chem, the model may

underestimate the concentrations of Br in the MBL. Indeed, the mean tropospheric Br and BrO concentrations are 0.03 and 0.13 ppt respectively, which are lower than 0.08 and 0.48 ppt in Schmidt et al. (2016) but are more consistent with 0.03 and 0.19 ppt simulated by Wang et al. (2021). The vertical distribution of zonal mean Br and BrO are shown in Fig. S2. The total Hg$^{II}$ deposition to the ocean is about 1600 Mg a$^{-1}$ higher than the net ocean emission in our simulation, which means that globally the ocean is a net atmospheric Hg sink (Soerensen et al., 2010b; Horowitz et al., 2017), especially in tropical oceans

with intensive precipitation.

We also compare the modeled Hg$^{II}$ wet deposition fluxes against observations over three regions in the NH: North America, East Asia, and West Europe. In North America, the model simulates a spatial distribution pattern that gradually increases from northwest to southeast in CONUS that is consistent well with the observations ($r = 0.80$, n = 147) (Figure 7). Previous models using Br as the sole oxidant (e.g., GEOS-Chem) failed to capture the maximum or underestimated the

magnitude of Hg$^{II}$ wet deposition along the coast of the Gulf of Mexico (Holmes et al., 2010; Amos et al., 2012; Horowitz et al., 2017). Our simulation with multiple oxidants agrees better with the observed maximum deposition fluxes (15–20 µg m$^{-2}$ a$^{-1}$), especially in the Florida Peninsula. The scatter plot of observed and modeled annual mean Hg$^{II}$ wet deposition flux is shown in Fig. S3. In a model comparison study, Travnikov et al. (2017) found that the OH/O$_3$ oxidation mechanisms showed a better agreement between modeled and observed wet deposition than Br oxidation mechanism, consistent with our study.

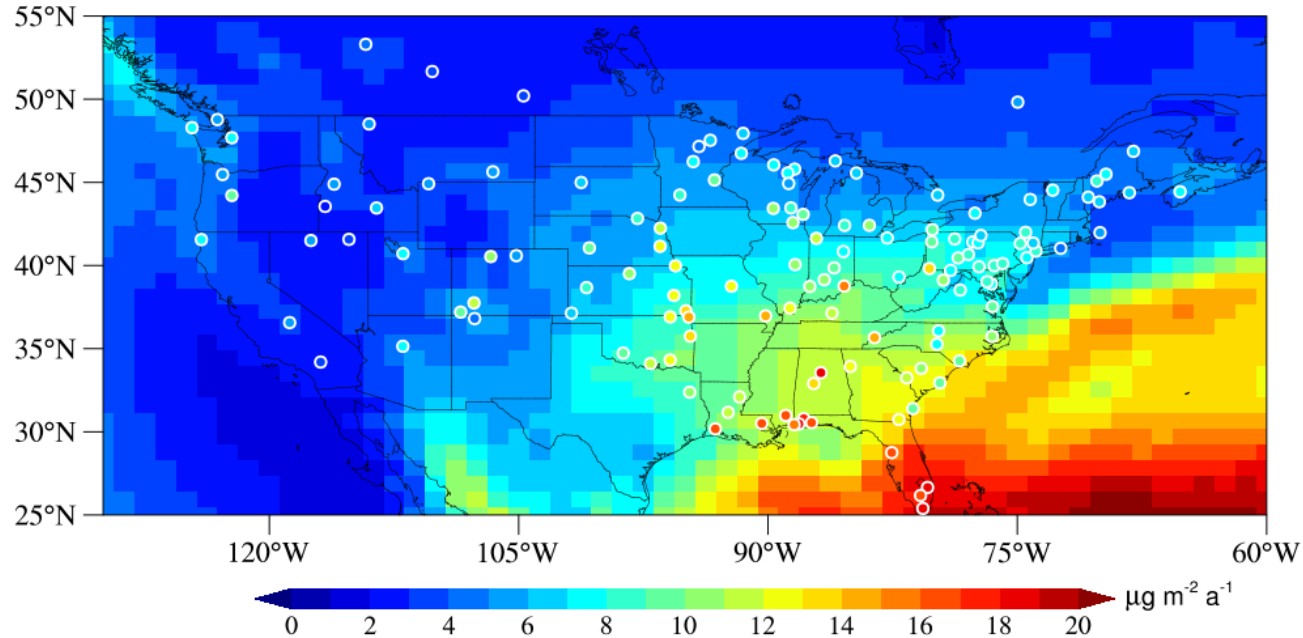

**Figure 7.** Comparison between modeled (background) and observed (circles) annual Hg$^{II}$ wet deposition fluxes over North America for 2011–2013. Annual observations are derived from the Mercury Deposition Network (MDN, National Atmospheric Deposition Program, http://nadp.slh.wisc.edu/MDN/) during 2009–2015.

Figure 8 shows the Hg$^{II}$ wet deposition over East Asia and West Europe, where the available observations are more limited (n = 9 and 7, respectively). In East Asia, the spatial distribution of Hg$^{II}$ wet deposition shows an increasing trend from inland northwest to coastal southeast, reflecting the pattern in anthropogenic emissions. The model captures the spatial distribution quite well ($r = 0.53$). Similar to North America, the model also simulates the maximum deposition flux over southeast China, probably due to the frequent convective precipitation there. More observations in this region are indeed 260 needed to confirm this prediction. Compared with the simulation by Horowitz et al. (2017), our model agrees better with limited observations in Chongqing and Nanjing, likely owing to the higher horizontal resolution of our model that improves the representation of emissions and precipitation (Zhang et al., 2012).

Compared to the above two regions, our simulation shows much smaller deposition flux and less spatial variability over West Europe. This pattern is in general agrees with the observations ($r = 0.74$), reflecting the fact that the wet deposition flux 265 is much higher at low latitudes than high latitudes (Figure 6c). The wet deposition flux is relatively high in the central regions (over Germany, United Kingdom, and Norway), where the Hg emissions are higher than other West European regions (Pacyna





et al., 2006; Ballabio et al., 2021). Apart from the impact of emissions, our model shows a high correlation between precipitation and wet deposition fluxes ($r$ = 0.72) in West Europe, indicating a strong influence of precipitation on Hg wet deposition flux over this region.

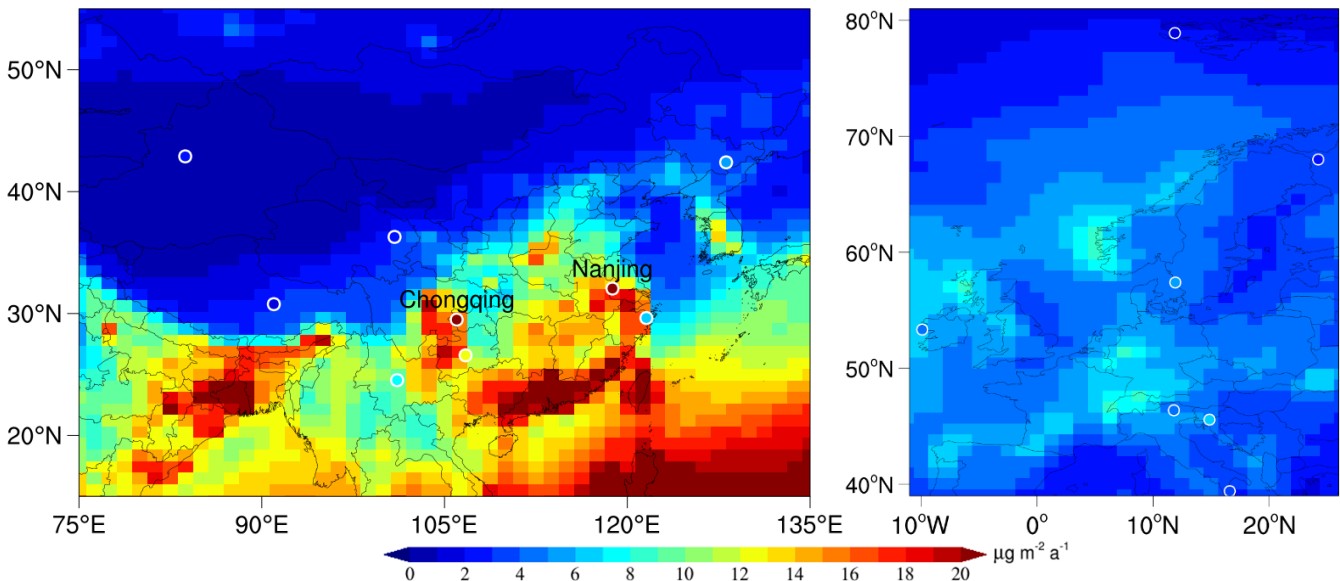


**Figure 8.** Same as Fig. 7, but for East Asia (left) and West Europe (right). The observations (circles) are derived from recent studies by Fu et al. (2015, 2016) and Sprovieri et al. (2017), respectively. For East Asia, only data with > 9 months data during 2009–2015 are included in the observations for comparison. For West Europe, the sites with sampling days greater than 60% in a year are taken as available observations.

## 4 Seasonal variations of atmospheric Hg


Figure 9 compares modeled and observed seasonal variations of TGM concentrations at ground-based sites in northern mid-latitudes (30°N–60°N, n = 43) and the SH (n = 5). In northern mid-latitudes, the observed TGM reaches the maximum in February, and are the lowest in September. In the SH, the observed TGM reaches the maximum in August and the minimum in March. In both hemispheres, the simulated results generally reproduce the observed seasonal variations. Notably, the

modeled amplitude of seasonal variability is greater in northern mid-latitudes than in the SH. Such a pattern has also been reported by several modeling studies (Selin et al., 2007; Holmes et al., 2010; De Simone et al., 2014; Horowitz et al., 2017), and is mainly attributed to four factors: anthropogenic and natural emissions, meteorological factors, $Hg^0$ oxidation rates, and $Hg^{II}$ deposition (Temme et al., 2007; Ebinghaus et al., 2011; Fu et al., 2015; Weigelt et al., 2015; Sprovieri et al., 2016). However, the role of $Hg^0$ uptake by vegetation is emphasized in a recent work (Jiskra et al., 2018), which shows a considerable

contribution to the seasonal variation of atmospheric $Hg^0$ distribution.



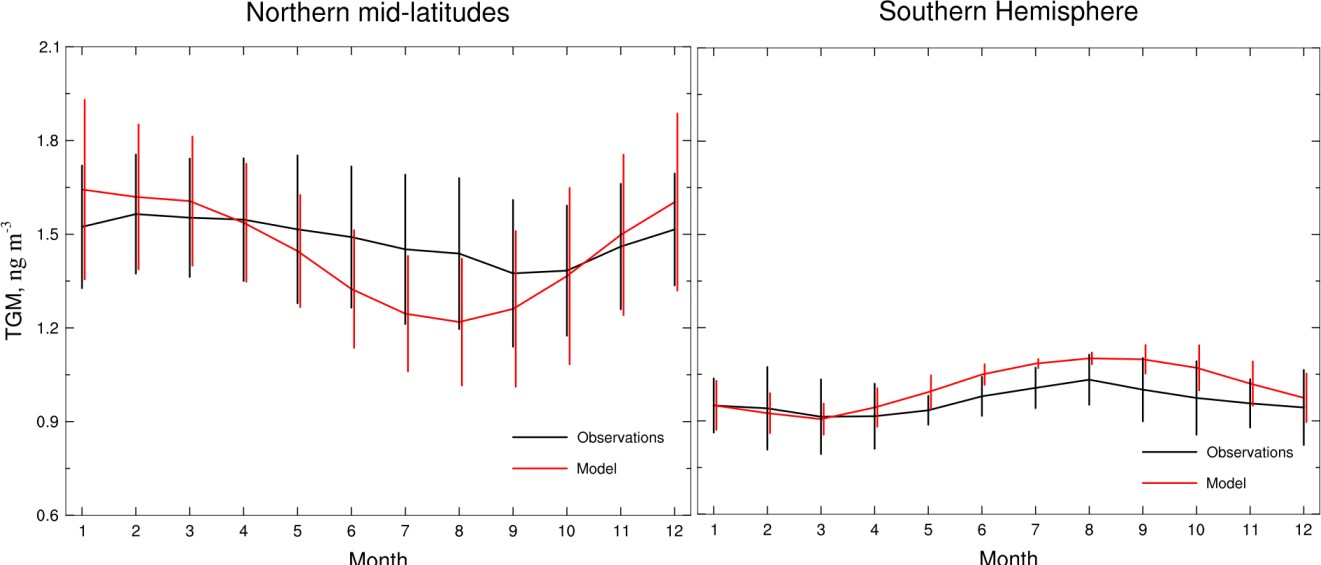

**Figure 9.** Seasonal variations of surface TGM concentrations in northern mid-latitudes (left) and the Southern Hemisphere (right). Observations are the average of the available monthly data from the ground-based sites in Fig. 3, indicated by the black line. The red line represents the results simulated by CAM6-Chem/Hg model. Error bar depicts the standard deviation of TGM concentrations.


We use the CAM6-Chem/Hg model to quantify the relative contribution of different factors to the seasonal variations of atmospheric Hg on a global scale. Since the anthropogenic emissions of Hg remain constant throughout the years in our simulation, we do not take this factor into account. Figure 10 shows the contributions of a variety of processes (dry and wet deposition fluxes for $Hg^0$ and $Hg^{II}$, as well as $Hg^0$ natural and legacy emissions) over land/ocean in the NH and SH for the year

2013. In the NH, $Hg^0$ dry deposition and $Hg^{II}$ dry/wet deposition have negative anomalies during November–April (bars and the red dashed line), leading to a continuous increase of atmospheric Hg levels (black line). Overall, $Hg^{II}$ dry/wet deposition (cyan and jacinth color bars, respectively) contributes the most (53%) to this increase, followed by $Hg^0$ dry deposition (deep blue color bars, 28%) and the natural/legacy emissions (orange color bars, 19%). During May–October, the removal processes of Hg from the atmosphere are higher than the annual average, which has caused a decrease in atmospheric Hg levels. Similarly,

$Hg^{II}$ dry/wet depositions are also the leading driving factor for this trend.

We find the changes of atmospheric Hg levels (black line) do not agree with the net source/sink terms for some months (e.g. October) (Figure 10a, red dashed line), likely due to the mixing time required for the source-sink processes to influence the hemispheric Hg pool. We thus split the global atmosphere into over land (Figure 10b) and over ocean (Figure 10c), respectively. The result over land is similar to that in the NH, but without significant lag between the net source/sink terms and the atmospheric Hg levels. We find that the contribution of $Hg^0$ dry deposition (40%) to the seasonal variations outweighs

that of $Hg^{II}$ wet depositions (32%) throughout the year, confirming previous thoughts (Wright et al., 2016; Jiskra et al., 2018; Obrist et al., 2021). However, we argue that the importance of $Hg^{II}$ total depositions also plays an important role, contributing





40% of the seasonality. The seasonal pattern of net source/sink terms over ocean (Figure 10c) is smoother compared to that on the land. The seasonal cycle is largely driven by the ocean re-emissions and Hg$^{II}$ wet depositions.

In the SH, we find the influence of Hg$^0$ dry deposition is much smaller than that of the NH, due to its much smaller land mass. Hg$^{II}$ depositions and Hg$^0$ natural/legacy emissions are the driving factors for the seasonal patterns of atmospheric Hg (Figure 10d). Notably, the amplitude of variation in atmospheric Hg level and net source/sink terms in the SH exceeds that of the NH. Hg$^{II}$ dry/wet deposition and Hg$^0$ dry deposition show negative anomalies during April–September, resulting in a sustained increase in atmospheric Hg levels. Slightly different from the NH, Hg$^{II}$ dry/wet deposition contributes the most (66%)

to this increase, followed by the natural/legacy emissions (28%) and Hg$^0$ dry deposition (6%). Not surprisingly, we find a much smaller seasonal amplitude for atmospheric Hg over SH land (Figure 10e) than NH land (Figure 10b) as a result of the lack of contribution from Hg$^0$ dry depositions. This result is supported by the observations in the Global Atmosphere Watch station at Cape Point, which shows a similar seasonal variation with our model (Sprovieri et al., 2016). The agreement between our model-derived net source/sink terms and atmospheric Hg levels is also the best (discrepancy < ±30 Mg) among the

land/ocean regions in the two hemispheres. Over the ocean, we find the Hg$^{II}$ wet deposition is the major driver for the seasonality (52%), followed by Hg$^{II}$ dry deposition (25%) and seawater Hg$^0$ evasion (23%).

        In conclusion, although the seasonal variations of atmospheric Hg in the NH and SH are similar, the relative contribution of different processes to the variations is different. It's a combination of different source and sink terms on a hemispheric scale, while there may be a dominant driving factor on a regional scale.

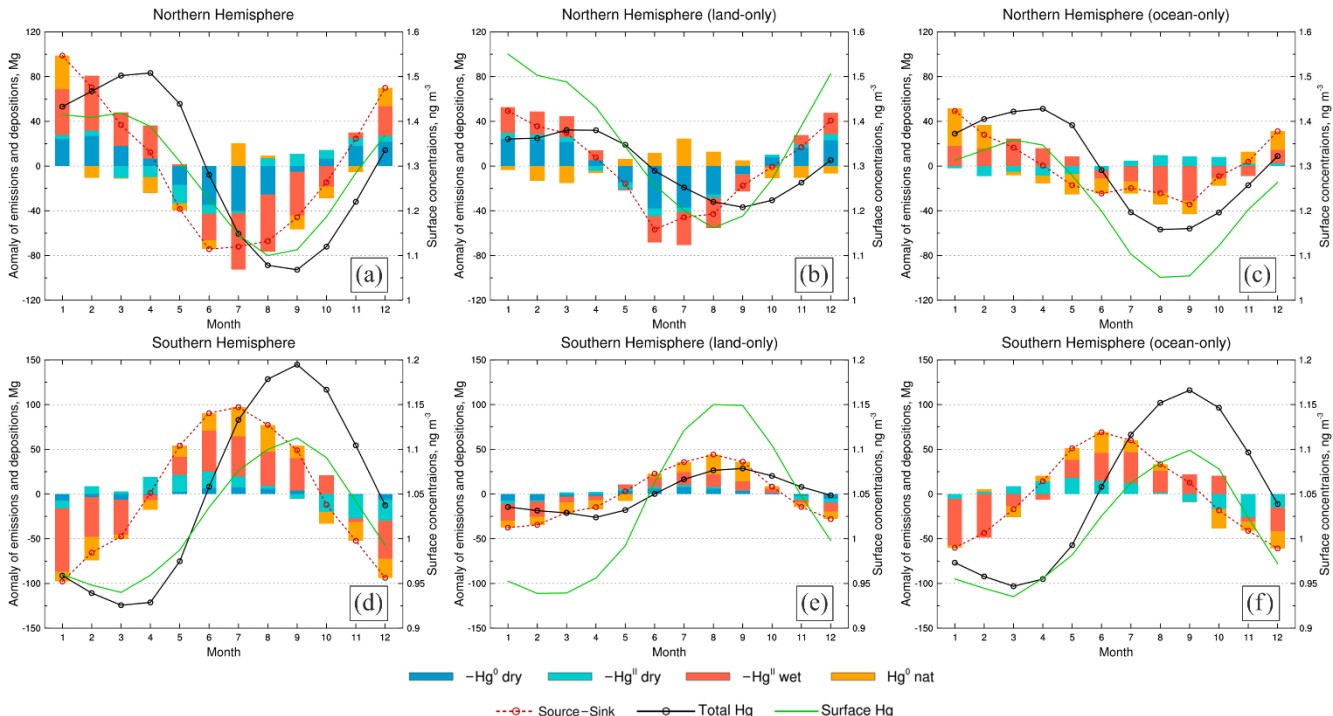




**Figure 10.** Seasonal variations of atmospheric Hg in (a) Northern Hemisphere, (b) Northern Hemisphere land areas only, (c) Northern Hemisphere ocean areas only, (d) Southern Hemisphere, (e) Southern Hemisphere land areas only, and (f) Southern Hemisphere ocean areas only. The bars represent the anomaly of different processes: deep blue for $Hg^0$ dry deposition, cyan for $Hg^{II}$ dry deposition, jacinth for $Hg^{II}$ wet deposition, and orange for $Hg^0$ natural/legacy emission. The first three are reversed for their signs as they are sinks of Hg. The anomaly of the difference between the source and the sink is shown as the red dashed line with circles, and the solid black line with circles is the anomaly of the total mass of atmospheric Hg. The solid green line represents the surface Hg concentrations.

## 5 Conclusions

The atmosphere is one of the most active parts of the global mercury cycle, as well as an important component for the mercury exchange between different components of the Earth system. Here we develop a new online global 3-D atmospheric mercury model (CAM6-Chem/Hg) and implement a chemical mechanism with multiple oxidation pathways of Hg. We use the online meteorological fields and built-in chemical species to drive the transport and physiochemical processes of Hg in the atmosphere. This online method reduces the uncertainties caused by the interpolation of external meteorological data and chemical species.

Two main oxidation pathways, Hg + Br and Hg + OH/$O_3$, are used together in the chemical mechanism. In our simulation, the lifetime of total gaseous mercury (TGM) against deposition is about 5.3 months, which is consistent with the observational constraints. The CAM6-Chem/Hg model also reproduced the ground-based observations and the interhemispheric gradient of TGM concentrations well. The global nominal 1° horizontal resolution used in CAM6-Chem/Hg can capture the distribution characteristics of TGM concentrations better on regional scales, especially in some high anthropogenic emission regions (e.g., East Asia). In particular, the CAM6-Chem/Hg calculates the $Hg^0$ dry deposition velocity online through coupling with the land component (i.e., CLM5) in CESM2, which can take the land-use types and seasonal variation into a good account. Comparing with Br as the sole oxidant, adding OH/$O_3$ as oxidants leads to a broader range and higher flux of $Hg^{II}$ wet deposition in low latitudes. The simulated $Hg^{II}$ wet deposition flux is consistent with the available observations and captures the magnitude of the maximum value in southeastern United States.

The CAM6-Chem/Hg reproduces the observed seasonal variations of TGM concentrations in northern mid-latitudes and the Southern Hemisphere, which show a trend of decrease in spring and summer and increase in autumn and winter. We use the model to quantify the processes affecting the seasonality of atmospheric Hg from the relationship between sources and sinks. On the hemispherical scale, the seasonal changes of deposition have an important influence on atmospheric Hg, followed by natural and re-emission sources. In the land region of the Northern Hemisphere, the contribution of $Hg^0$ dry deposition to seasonal variations of Hg outweighs that of $Hg^{II}$ wet deposition. The seasonal variations of atmospheric Hg are the result of multiple processes and have obvious regional characteristics. Research on $Hg^0$ dry deposition will improve our understanding of the seasonal cycle of atmospheric Hg.



## Code availability

The CESM2 model code is available at https://www.cesm.ucar.edu/models/cesm2/release_download.html (last access: 19 January 2022). The code used for implementing mercury into CAM6-Chem (CAM6-Chem/Hg v1.0) in this study is
permanently archived on Zenodo at https://doi.org/10.5281/zenodo.5877214 (last access: 19 January 2022).

## Author contribution

YZ and PZ conceived the idea and designed the model experiments. PZ and YZ modified the code of the model. PZ performed the simulations, conducted the analysis, and wrote the paper. YZ and PZ edited the paper.

## Competing interests

The authors declare that they have no conflict of interest.

## Acknowledgments

The CESM project is supported primarily by the National Science Foundation (NSF). We thank all the scientists, software engineers, and administrators who contributed to the development of CESM2. We would like to acknowledge and thank all the operators who collected the observations used in this work.

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
