# Peer review of "Earth system modeling of mercury using CESM2: part 1. atmospheric model CAM6-Chem/Hg v1.0"

_Geoscientific Model Development, 2021_

## Author Comment (AC1)

**Response to Referee #1:**

**We would like to thank the referee for the thoughtful and useful comments. We have provided our responses to the referee's comments below (in blue).**

**1. Comment:** Abstract, "but they have limited capacity in predicting the future"
It is better to simply explain the reasons why most global atmospheric mercury models have such limitations.

To clarify this, the sentence in line 6-8 was modified as:
"Most global atmospheric mercury models use offline and reanalyzed meteorological fields, which has the advantages of higher accuracy and lower computational cost compared to online models. However, these meteorological products need past and/or near real-time observational data and cannot predict the future."

**2. Comment:** "One advantage of our online model is that the concentrations of Hg oxidants are calculated online." What is the time resolution of the online Hg oxidants concentrations?

The time resolution of the online Hg oxidants concentrations is 1800 seconds, i.e. the native model time resolution of the model.

**3. Comment:** Aerosols concentrations significantly impact the species transformation of Hg in atmosphere? How are the performances of MAM4 and VBS in predicting the concentrations of aerosols as well as secondary organic aerosols?

Yes, the concentration of aerosols does significantly impact the species transformation of Hg. First, fine aerosol influences the gas-particle partitioning of $Hg^{II}$ (Amos et al., 2012. The related description is in line 118). Second, the uptake and deposition of $Hg^2$ by sea-salt aerosol is the dominant sink for $Hg^{II}$ in the MBL and the major source of Hg to the surface ocean (Holmes et al., 2010. The related description is in line 136).
The performances of MAM4 and VBS were clarified by modifying the sentence in line 75-80:
"Aerosols in CAM6-Chem are represented using the four-mode version of the Modal Aerosol Model (MAM4) (including sulfate, black carbon, primary organic matter, secondary organic aerosols, sea salt, and mineral dust), which significantly improves the representations of modeled black carbon and primary organic matter in many remote regions comparing to its previous version (Liu et al., 2016). Secondary organic aerosols are treated using a volatility basis set (VBS) scheme, which can alter and potentially improve organic aerosols' response to emissions and climate change (Tilmes et al., 2019)."

**4. Comment:** "The natural emissions are derived from the average of a 5-year simulation in GEOS-Chem, including geogenic, biomass burning, soil, snow, and vegetation emissions." Earth system has a natural advantage to couple mercury transfers between different spheres. It is better to calculate the natural emissions in CESM instead of using the results derived from GEOS-Chem directly.

Thank you for your valuable suggestion. CESM2 truly has a natural advantage in multi-spheres coupling, which is also the most important reason why we choose it as a tool for the simulation

of global Hg cycle. Atmospheric Hg is the first part of our model development, as stated in the title of our article. We are developing other components within CESM2 (e.g. ocean and land model for Hg) and would have more explicit natural emissions by then. For now, the natural emissions from GEOS-Chem are currently used as a space-holder.

We made it clear by modifying the sentences in line 106-107 as:

"Future work will have more explicit natural/legacy emissions and improve the calculation of the fluxes between different spheres by introducing Hg simulations into other models (e.g., CLM5 and POP2)."

**5. Comment:** "The best match with the available observations can be obtained by adjusting the photoreduction rate coefficient." Please compare the adjusted photoreduction rate coefficients with observations to verify the reasonability of the adjusted results.

Sorry for the misunderstanding. There are no direct observations of the photoreduction rate coefficients for $Hg^{II}$ in the liquid phase, which is often used as a tunning factor to adjust the model to match the global mean TGM surface observations (Horowitz et al., 2017; Shah et al., 2021). We modified the sentence in line 121-122 as:

"The best match with the available TGM surface observations and Hg wet deposition fluxes can be obtained by adjusting the photoreduction rate coefficient (Horowitz et al., 2017; Shah et al., 2021)."

**6. Comment:** "The representation of the main oxidants (e.g., O3 and OH) of $Hg^0$ have been greatly improved and are more comprehensive in the CAM6-Chem, comparing with its predecessors CESM or CCSM (Lamarque et al., 2012; Emmons et al., 2020)." Briefly describe the results please.

We clarified this by modifying the sentence in line 123-125 as:

"The representation of the main oxidants (e.g., $O_3$ and OH) of $Hg^0$ have been greatly improved and are more comprehensive in the CAM6-Chem compared with its predecessors CESM or CCSM. The simulated tropospheric $O_3$ also agrees better with ozonesonde observations worldwide (Lamarque et al., 2012; Emmons et al., 2020)."

**7. Comment:** Please introduce how legacy emissions and re-emissions are considered in the method part.

Thanks for pointing this out. The legacy emissions actually refer to the re-emissions of previously deposited Hg from land and ocean. So, they refer to the same thing in this context. Since the calculation of re-emissions relies on the land and ocean, the atmospheric Hg models traditionally treat re-emissions as constant input (Angot et al., 2018). We clarified this by adding the following sentences in line 99-101 in the method part:

"The natural emissions and re-emissions from previously deposited legacy emissions are derived from the average of a 5-year simulation in GEOS-Chem v11 (Horowitz et al., 2017), including geogenic, biomass burning, soil, ocean, snow, and vegetation emissions."

We also modified the sentence in line 153 as:

"The global total Hg emissions from all sources are about 7000 Mg $a^{-1}$, two-thirds of these are **natural/legacy** emissions."

**8. Comment:** "HOHgI and Hg(OH)2 formed by the oxidation of Hg0 by OH are relatively stable" cannot persuade the readers because the stability of oxidants does not necessarily relate to the significance of pathways. For example, HgBr· is not stable in atmosphere but it is a significantly mid product in the two-step Br oxidation process and Br· is argued as important oxidants. The oxidation pathways are significantly different from previous studies (e.g., Horowitz et al., 2017). Please compare with previous studies and explain the reasons as well as the reasonability of the results. In addition, does the dominant oxidation pathways differ across regions? Please briefly introduce the results?

We thank the reviewer for bringing it up. We made a mistake here: the $HOHg^I$ is not stable, but it can be oxidized by atmospheric radicals to form stable compounds HOHgY (Dibble 2020). The oxidation pathway of $Hg^0$ by OH has been used in many atmospheric Hg models and also can reproduce measured Hg concentrations and deposition fluxes (in line 34). To clarify this, we deleted this sentence in line 147 and added the following sentence in line 109-112: "Although the oxidation of $Hg^0$ by OH has been controversial, mainly due to the rapid thermal decomposition of its intermediate product $HgOH^I$ (Goodsite et al., 2004; Calvert and Lindberg, 2005), a recent study recalculated the HO-Hg bond energy and found that the OH-initiated oxidation of Hg plays an important role in polluted regions (Dibble et al., 2020)."

The dominant oxidation pathways differ across regions. We added a figure in SI (Figure S1) to show the spatial distribution of different oxidation pathways.

[Figure]

**Figure S1.** (Top) Annual zonal mean oxidation rates (molecules cm$^{-3}$ s$^{-1}$) of $Hg^0$ by OH (left), Br (middle), and $O_3$ (right). (Bottom) Global spatial distribution of annual mean $Hg^0$ oxidized (sum of all levels, Kg a$^{-1}$) by OH (left), Br (middle), and $O_3$ (right). Note that the scale is different for different oxidants.

We also added the following sentence in line 148-151 to describe Figure S1:
"And the dominant oxidation pathways differ across regions, which shows peaks in the mid-to-low latitude terrestrial regions for OH, while the oxidation by Br is stronger in the marine regions (Figure S1). This spatial distribution of different oxidation pathways is also consistent with the recent isotope data (Au Yang et al., 2021)."

**9. Comment:** Figure 4 Please present the uncertainty range of model results.

We added error bars in Figure 4 as suggested. The following sentence was added to the legend of this figure:
"The red line and shaded areas are the modeled means and standard deviations of 2011–2013, respectively."

[revised manuscript text omitted]

---

## Author Comment (AC2)

**Response to Referee #2:**

**We thank the reviewer for the acknowledgment and the helpful comments and suggestions. We have provided our responses to the referee's comments below (in blue).**

**1. Comment:** Shah et al., (2021) had a major revision to the GEOS-Chem mechanism described in Horowitz et al., (2017), how do the authors account for those changes? It is relevant to include a discussion on the mercury chemistry implemented in this study with Shah et al., 2021, which is the most updated mercury chemistry in a global model.

Shah et al., (2021) integrated the recent laboratory data of mercury and implemented a new chemical mechanism for atmospheric Hg into GEOS-Chem. The new mechanism in Shah et al., (2021) has more tracers and more complicated chemical processes compared to Horowitz et al., (2017). But it should be noted that the main purpose of developing CAM6-Chem/Hg is to test whether the online model could simulate the Hg cycle and for future prediction and coupling. The online simulation of CAM6-Chem/Hg is able to reproduce the observed surface and seasonal variations of TGM concentrations and the wet deposition fluxes reasonably well. Future model would benefit from both the updated Hg chemistry and online meteorology and chemistry.
We added the following sentence in line 360-362:
"Shah et al., (2021) presented a new chemical mechanism for atmospheric Hg in GEOS-Chem, and the CAM6-Chem/Hg presented here provides a convenient platform to integrate the updated Hg chemistry with online meteorology and atmospheric chemistry."

**2. Comment:** Why were not the emissions from Streets et al., (2015) used in the study? The Global Mercury Assessment 2018 used it in their report and in Shah et al (2021).

Since the natural/legacy emissions (and initial conditions) used in this study are from the result of GEOS-Chem (Horowitz et al., 2017), which is driven by the emissions from Zhang et al., (2016). Therefore, we used the same anthropogenic emission inventory instead of Streets et al., (2019) to keep consistency.
We clarified this by modifying the sentence in line 94:
"The anthropogenic emissions used in this study are consistent with Horowitz et al., (2017). They are based on an improved 2010…"

**3. Comment:** Which version of the GEOS-Chem model output was used for the natural emissions? Add the details in the manuscript.

Thanks for pointing it out. The natural and legacy emissions are derived from the GEOS-Chem v11 (Horowitz et al., 2017). We clarified this by adding the following sentences in line 99-101 in the method part:
"The natural emissions and re-emissions from previously deposited legacy emissions are derived from the average of a 5-year simulation in GEOS-Chem v11 (Horowitz et al., 2017), including geogenic, biomass burning, soil, ocean, snow, and vegetation emissions."

**4. Comment:** Is there a specific reason for choosing the simulation period 2011–2013?

Globally, observations of Hg are scarce and unfortunately becoming more scarce (e.g. many monitoring stations in AMNet and CAMNet are becoming inactive). We summarize the available observations of Hg to date (see Table S1 in SI) and found the data around 2011–2015 are the most abundant. So we choose the simulation period 2011–2013 to better constrain our model.

We clarified this by modifying the sentence in line 141:

"…the following three years (i.e., 2011–2013, **with abundant observations during the past decades**) for analysis (unless explicitly stated)"

**References**

Horowitz, H. M., Jacob, D. J., Zhang, Y., Dibble, T. S., Slemr, F., Amos, H. M., Schmidt, J. A., Corbitt, E. S., Marais, E. A., and Sunderland, E. M.: A new mechanism for atmospheric mercury redox chemistry: implications for the global mercury budget, Atmos. Chem. Phys., 17, 6353-6371, 10.5194/acp-17-6353-2017, 2017.

Shah, V., Jacob, D. J., Thackray, C. P., Wang, X., Sunderland, E. M., Dibble, T. S., Saiz-Lopez, A., Černušák, I., Kellö, V., Castro, P. J., Wu, R., and Wang, C.: Improved Mechanistic Model of the Atmospheric Redox Chemistry of Mercury, Environ. Sci. Technol., 10.1021/acs.est.1c03160, 2021.

Streets, D. G., Horowitz, H. M., Lu, Z., Levin, L., Thackray, C. P., Sunderland, E. M., and Argonne National Lab. ANL, A. I. U. S.: Five hundred years of anthropogenic mercury: spatial and temporal release profiles, Environ. Res. Lett., 14, 84004, 10.1088/1748-9326/ab281f, 2019.

Zhang, Y., Jacob, D. J., Horowitz, H. M., Chen, L., Amos, H. M., Krabbenhoft, D. P., Slemr, F., St. Louis, V. L., and Sunderland, E. M.: Observed decrease in atmospheric mercury explained by global decline in anthropogenic emissions, Proceedings of the National Academy of Sciences, 113, 526-531, 10.1073/pnas.1516312113, 2016.